# Single-Cell Approach Reveals Intercellular Heterogeneity in Phage-Producing Capacities

Sherin Kannoly,[a] Gabriella Oken,[a] Jonathan Shadan,[a] David Musheyev,[a] Kevin Singh,[a] Abhyudai Singh,[b] John J. Dennehy[a,c]

[a]Biology Department, Queens College of The City University of New York, New York, New York, USA
[b]Department of Electrical and Computer Engineering, University of Delaware, Newark, Delaware, USA
[c]The Graduate Center of The City University of New York, New York, New York, USA

**ABSTRACT** Bacteriophage burst size is the average number of phage virions released from infected bacterial cells, and its magnitude depends on the duration of an intracellular progeny accumulation phase. Burst size is often measured at the population level, not the single-cell level, and consequently, statistical moments are not commonly available. In this study, we estimated the bacteriophage lambda ($\lambda$) single-cell burst size mean and variance following different intracellular accumulation period durations by employing *Escherichia coli* lysogens bearing lysis-deficient $\lambda$ prophages. Single lysogens can be isolated and chemically lysed at desired times following prophage induction to quantify progeny intracellular accumulation within individual cells. Our data showed that $\lambda$ phage burst size initially increased exponentially with increased lysis time (i.e., period between induction and chemical lysis) and then saturated at longer lysis times. We also demonstrated that cell-to-cell variation, or "noise," in lysis timing did not contribute significantly to burst size noise. The burst size noise remained constant with increasing mean burst size. The most likely explanation for the experimentally observed constant burst size noise was that cell-to-cell differences in burst size originated from intercellular heterogeneity in cellular capacities to produce phages. The mean burst size measured at different lysis times was positively correlated to cell volume, which may determine the cellular phage production capacity. However, experiments controlling for cell size indicated that there are other factors in addition to cell size that determine this cellular capacity.

**IMPORTANCE** Phages produce offspring by hijacking a cell's replicative machinery. Previously, it was noted that the variation in the number of phages produced by single infected cells far exceeded cell size variation. It was hypothesized that this variation is a consequence of variation in the timing of host cell lysis. Here, we show that cell-to-cell variation in lysis timing does not significantly contribute to the burst size variation. We suggest that the constant burst size variation across different host lysis times results from cell-to-cell differences in capacity to produce phages. We found that the mean burst size measured at different lysis times was positively correlated to cell volume, which may determine the cellular phage production capacity. However, experiments controlling for cell size indicated that there are other factors in addition to cell size that determine this cellular capacity.

**KEYWORDS** burst size, lysis time, noise, phage life history, single cell

Address correspondence to Abhyudai Singh, absingh@udel.edu, or John J. Dennehy, john.dennehy@qc.cuny.edu.

The authors declare no conflict of interest.

As part of their life cycle, most bacteriophages (phages) lyse host bacterial cells after assembling progeny virions. Lysis time, which is defined as time elapsed between host cell infection and lysis, and burst size, which is defined as the mean number of virions produced per infected host cell, are therefore key traits that affect phage evolutionary fitness and are analogous to the generation time and fecundity of higher organisms (1). However, this analogy breaks down in that burst size is positively

correlated to lysis time—a longer lysis time will result in a higher burst size (2–5). Intriguingly, lysis time, and hence burst size, is a highly malleable trait as single-nucleotide substitutions in the genetic sequence of the lysis time regulator protein holin can result in dramatic differences in lysis time (6, 7). Thus, lysis timing phenotypes ranging from extremely short to extremely long are easily accessible in the sequence space available to natural selection (6, 7). It follows, then, that phages can, in principle, adapt to local ecological and environmental conditions with fine-grained temporal resolution such that they can achieve a lysis time optimum where reproductive output is maximized (6, 8–10). As such, variation in lysis time and burst size can significantly impact phage and host population dynamics (6, 8, 10, 11).

Despite the central importance of these bacteriophage life history traits, the precise relationship between lysis time and burst size, and how variation in the former affects the latter, remains opaque. For instance, it is not clear whether the correlation between lysis time and burst size is linear or nonlinear (2, 6, 12–15). While some estimates have suggested that the relationship is linear, it should be noted that some of these estimates were taken from bulk populations and are therefore somewhat imprecise. Nor is it clear how the cell-to-cell variation or "noise" in lysis timing affects noise in burst size. Ever since the work of Delbrück, it has been known that variation in burst size far exceeds that of lysis time and cell size (16). A key question is whether burst size noise increases with increasing lysis time, but no study has assayed burst size noise in isogenic phages that vary with respect to lysis time. Understanding these effects will permit more accurate modeling of phage and bacterial population dynamics.

Here, we used the bacteriophage lambda ($\lambda$) as a model system to quantify phage single-cell burst sizes and the burst size statistical moments across a range of lysis times (6, 17–19). In a previous study, we used a panel of phage $\lambda$ holin mutants that varied in their lysis times to show that the noise in lysis timing took a concave-up shape with respect to lysis time, suggesting an optimal lysis time that minimizes noise (17). A theoretical approach further suggested that the noise in lysis timing is minimized when both holin and its antagonist, antiholin, are expressed at an optimal ratio (20). In a follow-up study, we elucidated the biological significance of minimizing the noise in lysis timing and demonstrated that there exists a range of optimal lysis times where phage fitness is maximized in a quasi-continuous culture (8).

Encouraged by these results, we further explored how burst size noise varied across a range of lysis times. The conventional method for estimating burst size— the one-step growth curve developed by Ellis and Delbrück—generates an average burst size by dividing the total number of phage progeny produced during a single round of infection by the total number of infected cells (21). Therefore, this method is unable to provide estimates of cell-to-cell variation (noise) in burst size. To measure cell-to-cell variation in burst size, single infected cells must be isolated and lysed so that their progeny may be counted, e.g., as in the method devised by Burnet (22). In this study, we modified Burnet's method to estimate phage $\lambda$ burst size at a single-cell level and to observe how the burst size changed with respect to changes in lysis time. Our method employed a variant of the temperature-sensitive phage $\lambda$ cI857 strain, in which the introduction of an amber mutation into the holin gene results in a truncated holin protein that is incapable of initiating cell lysis (23, 24). After *Escherichia coli* lysogens bearing this mutated $\lambda$ strain are thermally induced, cells do not divide, and phage progeny will accumulate intracellularly until the cell is lysed by external means. Induced cultures of cells bearing lysis-deficient phages were diluted and aliquoted to wells of a 96-well plate such that each well contained on average 0.25 cells (Fig. 1). After allowing for a range of different incubation times, cells were artificially lysed using chloroform, and the resulting progeny were quantified by plating on *E. coli* cells that expressed functional holin in *trans* and thus could complement the lack of functional holin produced by the

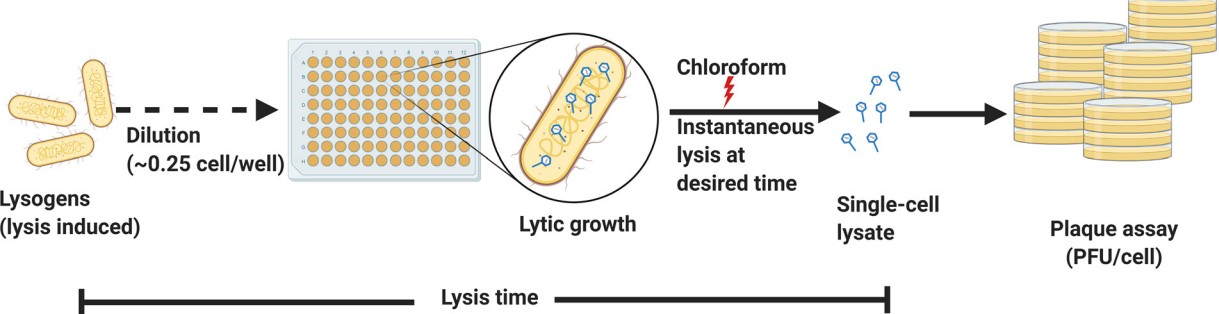

**FIG 1** Single-cell lysates can be used to estimate burst size. An induced culture of *E. coli* lysogen with lysis-deficient λ phage was diluted and aliquoted into a 96-well plate to give on average ≈0.25 cells/well, which minimized the probability of having multiple cells per well. The lytic cycle was allowed to proceed until the cells were chemically lysed, which resulted in single-cell lysates in some wells. The contents of each well were used in plaque assays to estimate burst size.

phage itself. In this manner, we were able to quantify single-cell burst size and distribution for a wide range of lysis times.

## RESULTS AND DISCUSSION

To estimate single-cell burst sizes, we employed a lysis-deficient λ lysogen which can be thermally induced to initiate the lytic cycle. In studies using microscopy, we have observed that the lysogenic cells do not divide following heat induction. This allows the lytic cycle to proceed unhindered. Thus, cells are "phage factories" that assemble phages until they are chemically lysed using chloroform (Fig. 1). Adsorption of the released phages to host LamB receptors was minimized by supplementing the growth medium with glucose, which represses LamB expression. The released phages were lysis deficient; thus, an *E. coli* host that expresses functional holin in *trans* was employed for plaque assays. This system enabled us to estimate phage λ single-cell burst sizes at different lysis times.

Induced cells were diluted such that each well of a 96-well plate received on average 0.25 cells, to minimize the probability that wells received multiple cells. If performed accurately, each plate will contain 74 wells with no cells. Plaque assays using the contents of such wells will produce no plaques. As a result, to calculate the burst sizes for at least 100 cells, around 500 plaque assays were performed for each time point. However, batch-to-batch variation resulted in empty wells ranging from 74 to 52, suggesting that the wells received on average 0.25 to 0.61 cells/well. Based on this range and assuming a Poisson distribution, we predicted that the number of wells that received 2 cells ranged from 2% to 10%. Frequencies of wells receiving three or more cells were negligible. The burst sizes obtained from wells with multiple cells should lie at the far end of the burst size distribution and may explain the right skewness of these distributions (Fig. 2, upper right panel). The estimated skewness of burst size distributions ranged from 0.5 to 2 (moments package in R), suggesting a moderate (0.5 to 1) to high (>1) positive skewness, which confirmed the presence of multiple cells in some wells. These wells were outliers and resulted in an overestimation of the mean burst size. To study the effect of outliers, we discarded the data that were more than twice the observed mean. In all cases, this resulted in less than 10% reduction in sample size. This also reduced the skewness of all distributions, and the means were recalculated using the trimmed data. The recalculated mean burst sizes for most distributions showed a reduction of 15% or less, except in two cases where the reduction was 20% (see Table S1 in the supplemental material).

When plotted against time, the mean burst sizes for approximately 100 cells was best represented by a sigmoidal curve that plateaued at about 3 h (Fig. 2). Note that all data shown in Fig. 2 are the original data that included the outliers. Thus, our data showed that the viral burst size initially increased exponentially with the lysis time and

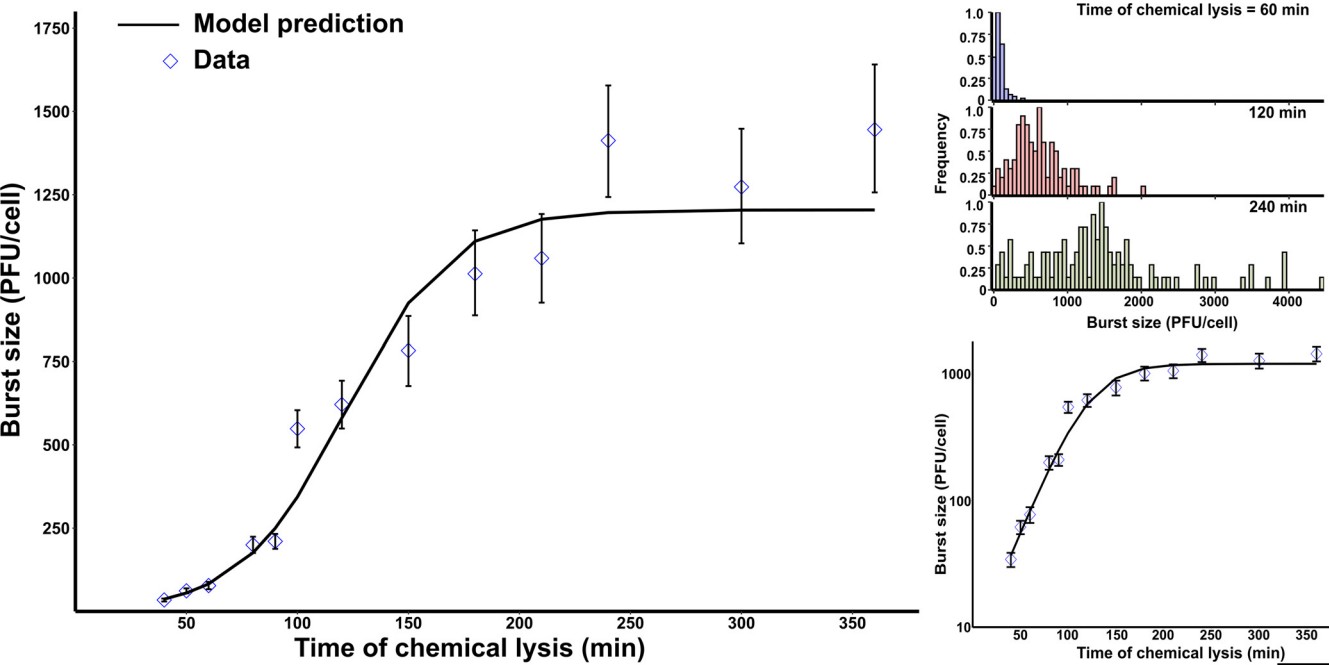

**FIG 2** Burst size first accelerated exponentially and then saturated at longer lysis times. After inducing the lytic cycle, single cells of lysogenic *E. coli* with lysis-deficient λ phage were chemically lysed at regular intervals. (Left) Mean burst size calculated for approximately 100 cells for each lysis time. The error bars represent 95% confidence intervals (CIs) after bootstrapping (1,000 replicates), and the black line represents the model fit (equation 1). (Upper right) Burst size distributions for three lysis times. (Bottom right) Exponential increase in burst size when data were plotted on the *y* axis using a log scale.

then saturated at longer lysis times (Fig. 2, left and bottom right panels). Motivated by these data, we phenomenologically modeled the burst size (BS) as a function of lysis time (LT) by equation 1:

$$\mathrm{BS} = k_{\max} \frac{e^{r(\mathrm{LT}-D)} - 1}{e^{r(\mathrm{LT}_{50}-D)} + e^{r(\mathrm{LT}-D)} - 2}, \quad \mathrm{LT} \geq D \qquad (1)$$

where $D$ is a time delay such that BS = 0 for LT ≤ $D$. $k_{\max}$ represents the maximum BS that can be characterized as the cellular capacity to produce phages. $\mathrm{LT}_{50}$ is the lysis time where BS = $k_{\max}/2$ and $r$ is the exponential growth rate. Fitting equation 1 to data, we estimated that $D \approx 25$ min, $r \approx 0.027$ min$^{-1}$, $\mathrm{LT}_{50} \approx 119$ min, and $k_{\max} \approx 1,430$ PFU/cell (Fig. 2).

To address whether the chloroform treatment had any effect on phage viability, we exposed phage lysates to the same experimental procedures and compared the resulting titers to those for a chloroform-free control group. The results indicated that the experimental conditions (including chloroform exposure) reduced the phage numbers by 19% (Fig. S1). Our burst size estimations therefore were likely underestimated by 19%. This means that the burst size curve in Fig. 2 (left and bottom right panels) would be shifted slightly higher along the *y* axis with a higher $k_{\max}$ value. The experimental procedures were performed under the assumption that the holin protein acts as a lysis clock that is only involved in the lysis process and does not interfere or regulate the phage protein expression or the assembly process. To check if holin has any regulatory role in phage assembly, we compared the burst sizes of naturally lysing strains to the that of the holin-deficient strain lysed using chloroform. We found no significant differences in the burst sizes obtained from natural or chemical lysis (Fig. S2).

It is important to understand why the burst size saturates at longer lysis times. First, we asked if the lack of nutrients was a limiting factor for continuous phage production. A previous study estimated that a T4 phage infection with a burst size of 200 required 30% of the host energy supply (25). Being smaller than T4 phage, phage λ may be assumed to have similar or lower energy requirements. *E. coli* growing in a complex

and rich medium such as LB has a carrying capacity of $\approx 5 \times 10^9$ CFU/mL, and so a single metabolically active cell in 200 $\mu$L of LB can theoretically saturate the medium with a total of about 1 billion cells (26). Yet, the burst size in a single cell only reached a maximum of 1,000 phage, suggesting that assembly and maturation of virions are not limited by nutrients required to sustain cell metabolism.

Another possibility is that phage production is limited by the available intracellular space. The estimated volume of an *E. coli* cell is $\approx 4.4$ $\mu$m³ (27) and that of T4 phage is $\approx 3.54 \times 10^{-4}$ $\mu$m³ (28), which suggests that an *E. coli* cell can theoretically harbor $\approx 10^4$ T4 virions. However, it has been estimated that 18% of the cell volume is occupied by proteins (17%) and DNA (1%) (29, 30). Ribosomes occupy 10% of the cellular volume, which leaves less than 8% out of the 18% for phage accumulation. If we assume that this 8% of cellular volume (0.352 $\mu$m³) can be entirely occupied by T4 virions, then the maximum T4 burst size can reach approximately 1,000 virions. However, we observed increases in cell volumes after induction of the lytic cycle (Fig. S3). As burst size is positively correlated to the cell volume (Fig. S4), we suspected that the cellular capacity ($k_{max}$) for phage production is mostly the result of the available intracellular space. However, the rate of increase in cell volume slowed toward the later part of the lytic cycle (Fig. S3). The theoretically available intracellular space may limit phage production as the burst size approaches saturation at around 1,000 virions in 3 h. This result suggested that cell volume might impose spatial constraints in limiting the number of phages that can be assembled, especially at longer lysis times. Thus, the differences in cellular capacity to produce phages might be explained by the observed differences in cell size.

To test this idea, we used fluorescence assisted cell sorting (FACS) to estimate the burst size of similarly sized cells. In these experiments, a FACS Aria instrument was used to sort cells with similar light scattering properties into wells of a 96-well plate. If cell size is the sole determinant of cellular capacity, then the cell-to-cell variation in burst size estimated from cells of similar size would approach zero. To quantify the cell-to-cell variation or "noise" in burst size, we estimated noise as a unitless metric, the coefficient of variation (CV, standard deviation divided by the mean). Our data showed that the burst size noise (CV) estimated from similar-sized cells with a mean burst size of 82.62 was 0.42 (Fig. S5). These numbers are comparable to burst size measurements of the same strain by standard techniques reported in another study, where mean burst size was 88.5 with a CV of 0.80 (6). However, phage burst size in this study was estimated in bulk from a population of cells with a wide range of naturally occurring cell sizes, which might explain the higher noise compared to single-cell estimations. The lower CV for the FACS-sorted cells probably reflects the narrower range of cell sizes for sorted cells compared to those found in ordinary *E. coli* broth cultures. This finding suggests that, in addition to cell size, there are other factors that determine the cellular capacity to produce phages.

What other factors might influence phage production? Being obligate intracellular parasites, phages use the cellular machinery for genome replication and assembly of virions. The concentrations of ribosomes and other proteins, which play key roles in the translation and replication processes, depend on the physiological and metabolic states of a given cell (31). For example, the distribution of ribosomes in newly divided daughter cells is unequal, and the number of ribosomes begins to increase halfway through the cell cycle and peaks close to cell division (32). For our burst size estimations, we induced exponentially growing cells that were not synchronized with respect to their cell cycles. Cells at different stages of growth may present with distinct environments affecting the phage life cycle. Therefore, it is not hard to imagine that the intracellular milieu of a cell would have profound effects on overall virion production (31).

A study using T4 phages showed that phage productivity in cells close to cell division was almost 3 times greater than the productivity of newly divided daughter cells (15). That study also found that the intracellular RNA levels and not the DNA

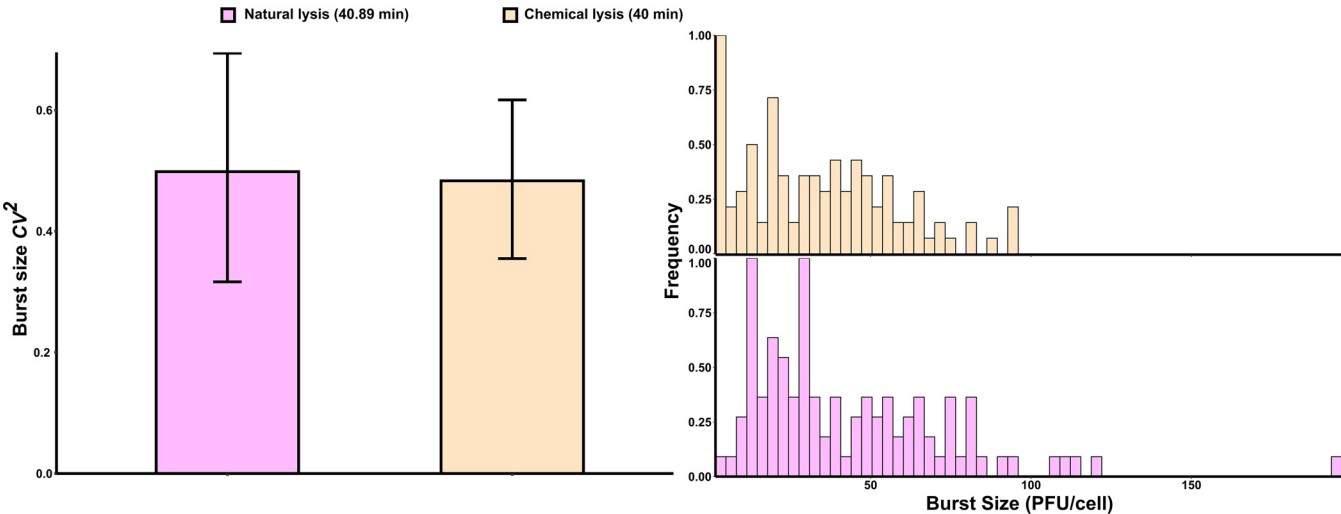

**FIG 3** Chemically and naturally lysed cells showed a similar cell-to-cell variation (noise) in burst size ($CV^2$). Approximately 100 individual cells of an *E. coli* lysogen with lysis-deficient $\lambda$ phage were induced and chemically lysed at 40 min to estimate the noise in burst size as quantified using the coefficient of variation squared ($CV^2$). This noise was compared to that of lysogenic cells ($n \approx 100$) harboring a mutant holin that shortened the mean lysis time to 40.89 min (left). The normalized distribution of single-cell burst sizes ($\approx$100 cells) for both lysogens is shown on the right. Error bars, 95% CIs after bootstrapping with 1,000 replicates.

levels were strongly correlated to phage productivity. It is interesting to note that, following induction of the lytic cycle, cytokinesis was blocked by $\lambda$ via ZipA-dependent inhibition of FtsZ (33). It is possible that as the lytic cycle proceeds, the increased availability of intracellular resources accelerates phage production. It would be interesting to explore how cell cycle synchronization, which can be controlled with serine hydroxymate, impacts burst size and lysis time noise (34). In addition, experiments where relative concentrations of key cellular proteins are simultaneously quantified along with phage burst size may provide insights into the factors affecting phage production capacity.

Next, we investigated if cell-to-cell differences in the lysis time affected variation in the viral burst size. Previous studies using the lysis-deficient lysogen had demonstrated the lytic effect of chloroform on induced cells. When added to an induced culture, chloroform permeabilizes the inner membrane of these lysogens, which results in instant loss of turbidity (35, 36). Lysis events mediated by functional holin will show cell-to-cell variation in lysis timing. However, chloroform-induced lysis will minimize such variations, especially in our experimental setup where a well most likely contains a single cell (Fig. 1). The use of a multichannel pipette to quickly add chloroform to each well minimized the variation in lysis timing of single cells. This instantaneous lysis ensured negligible contribution of lysis time variation to the burst size variation. We compared the burst size noise in chloroform-lysed cells to that of a naturally lysing holin mutant (see strain JJD251 data in Table 1 of Kannoly et al., 2020 [17]) with a mean lysis time of $\approx$40 min (Fig. 3 and Fig. S6, lysis time of $\approx$60 min). If the noise in lysis timing significantly contributes to the noise in burst size, then the burst size CV of the naturally lysing holin mutant would be significantly higher than that of the chemically lysed mutant. We used the R package cvequality version 0.1.3 (37) to test for significant differences in the CVs. The noise levels in burst size estimated from natural and chemical lyses were not significantly different (asymptotic test, $P = 0.81$; modified signed-likelihood ratio test, $P = 0.82$), suggesting that the noise in lysis timing did not significantly contribute to the noise in burst size.

Using the burst size estimations, we further calculated the noise in burst size at different lysis times. After removing the outliers to reduce skewness, the estimated noise in burst size was slightly reduced (Fig. S7 and Table S1). Interestingly, burst size noise appeared to remain fairly constant with increasing mean burst size (Fig. 4). Recalculating the burst size noise after removing the outliers also produced similar results (Fig. S7 and

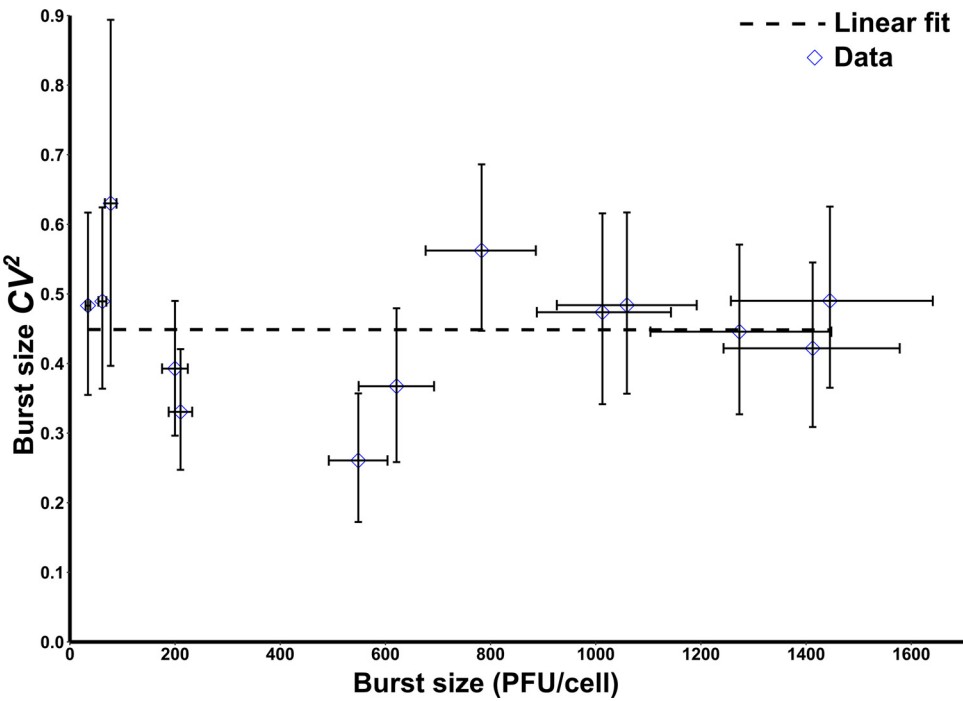

**FIG 4** Cell-to-cell variation (noise) in burst size remained constant with increasing mean burst size. Noise ($CV^2$) in burst size is shown plotted against mean burst sizes estimated for a range of lysis times. Each point represents the mean burst size estimated from $\approx 100$ cells. The dotted line is a linear fit of the data. Error bars, 95% CIs after bootstrapping (1,000 replicates).

Table S1). Inspired by these data, we attempted to explain the constant burst size noise using the parameters in equation 1. To model the cell-to-cell variation in burst size (BS), we first wrote equation 1 as follows:

$$BS = f(LT, k_{max}, LT_{50}, r), \tag{2}$$

and then considering small intercellular differences in parameters $k_{max}$, $LT_{50}$, and $r$ around their respective means, $\hat{k}_{max}$, $\widehat{LT}_{50}$, and $\hat{r}$, we expanded the right side of equation 2 as follows:

$$BS = \widehat{BS} + \frac{\partial f}{\partial k_{max}}\left(k_{max} - \hat{k}_{max}\right) + \frac{\partial f}{\partial LT_{50}}\left(LT_{50} - \widehat{LT}_{50}\right) + \frac{\partial f}{\partial r}\left(r - \hat{r}\right) \tag{3}$$

where the mean burst size $\widehat{BS}$ is given by

$$\widehat{BS} = f\left(LT, \hat{k}_{max}, \widehat{LT}_{50}, \hat{r}\right)$$

Rearranging terms in equation 3 we obtain

$$\frac{BS - \widehat{BS}}{\widehat{BS}} = \frac{\hat{k}_{max}}{f}\frac{\partial f}{\partial k_{max}}\frac{k_{max} - \hat{k}_{max}}{\hat{k}_{max}} + \frac{\widehat{LT}_{50}}{f}\frac{\partial f}{\partial LT_{50}}\frac{LT_{50} - \widehat{LT}_{50}}{\widehat{LT}_{50}} + \frac{\hat{r}}{f}\frac{\partial f}{\partial r}\frac{r - \hat{r}}{\hat{r}}. \tag{4}$$

By squaring both sides of equation 4 and taking the expected value, we obtained the following expression for the $CV^2$ of the viral burst size:

$$CV^2_{BS} = \left(S^f_{k_{max}}\right)^2 CV^2_{k_{max}} + \left(S^f_{LT_{50}}\right)^2 CV^2_{LT_{50}} + \left(S^f_r\right)^2 \frac{\partial f}{\partial r} CV^2_r \tag{5}$$

where $CV^2_{k_{max}}$, $CV^2_{LT_{50}}$, and $CV^2_r$ are the cell-to-cell variations in parameters $k_{max}$, $LT_{50}$, and $r$, respectively, as quantified by their coefficients of variation squared. Equation 5 is

characterized by the following dimensionless log sensitivities of function $f$ with respect to $k_{max}$, $LT_{50}$, and $r$.

$$S_{k_{max}}^{f} = \frac{k_{max}}{f}\frac{\partial f}{\partial k_{max}} = 1$$

$$S_{LT_{50}}^{f} = \frac{\widehat{LT}_{50}}{f}\frac{\partial f}{\partial LT_{50}}$$

$$S_{r}^{f} = \frac{\hat{r}}{f}\frac{\partial f}{\partial r}CV_{r}^{2}$$

Note that the sensitivity of $f$ with respect to $k_{max}$ is 1 and is invariant of the lysis time. Thus, from equation 5, cell-to-cell differences in capacity $k_{max}$ results in a constant $CV_{BS}^{2}$, as seen in the data (Fig. 4). Our analysis of $S_{LT_{50}}^{f}$ and $S_{r}^{f}$ revealed that they can vary nonmonotonically with the lysis time but always approached 0 for longer lysis times. Thus, if differences in $LT_{50}$ and $r$ were the dominant drivers of viral burst size fluctuations, then $CV_{BS}^{2}$ would be predicted to decrease to 0 with increasing mean burst size. In summary, the most parsimonious explanation for the experimentally observed constant $CV_{BS}^{2}$ in Fig. 4 is that cell-to-cell differences in burst size originated from differences in cellular capacity to produce phages ($k_{max}$). In other words, the constant burst size noise points toward intercellular heterogeneity in cellular capacities to produce phages (38–40).

In this study, we showed that it was possible to simultaneously estimate phage burst size and variation in burst size at different lysis times. The classic one-step growth curve of phage biology can only estimate lysis time and burst size values averaged across a phage population; thus, it cannot be used to estimate the cell-to-cell variation in burst sizes. Burnet invented the first method for single-cell burst size estimation by diluting a phage-infected suspension of bacterial cells into small aliquots such that each aliquot contained on average less than 1 infected bacterium (22). This strategy ensured that only a small fraction of aliquots would contain more than 1 infected bacterium. After incubation for sufficient time to allow lysis of all bacteria, plaque assays revealed the distribution of burst sizes. This method was later improved to estimate the burst size distribution in a larger sample size (16). These early methods used free phages to initiate infection by allowing adsorption onto growing cells, and they required quick dilution and distribution of cells before the first burst occurred. This introduced variation resulting from the time required for phages to adsorb onto the host cells and inject their genomes. By contrast, our method uses a lysis-deficient lysogen, which allowed us to induce the lytic cycle simultaneously in all cells as well as to lyse the cells at desired times. The chemical lysis of several individual cells could be triggered at almost the same time, which reduced the cell-to-cell variation in lysis times observed in naturally lysing lysogens. This in turn minimized the effects of lysis time variation on burst size variation. Thus, the method is an improvement over previous attempts where nonlysogenic cells were infected with free phages, which was followed by dilution, to obtain single-cell estimates of burst sizes. Our method allowed more accurate estimations of burst size mean and its distribution at different lysis times. Using this method, we observed that the noise in lysis timing did not significantly contribute to the noise in burst size and that the burst size noise remained constant across different lysis times. The most likely explanation for the experimentally observed constant burst size noise is that cell-to-cell differences in burst size originate from intercellular heterogeneity in cellular capacities ($k_{max}$) to produce phages.

Like most cellular processes, the noise in lysis time and burst size estimations arises from stochastic expression of genes involved in these processes. However, the noise in burst size is at least an order of magnitude greater than the noise in lysis time (this study and reference 17). Although lysis occurs due to the action of four proteins of the lambda lysis cassette (S, R, Rz, and Rz1), it is triggered mainly as a result of holin (S)

**TABLE 1** Bacterial strains

| Strain | Genotype | Source |
|---|---|---|
| CGSC 6152[a] | *E. coli* MC4100 ($\lambda^-$) | Casadaban (41) |
| JJD14 | *E. coli* MC4100 ($\lambda$ cI857 $S_{am}7$) $\lambda$III | Wang (6) |
| JJD459 | *E. coli* MC4100 + pS$_{wt}$ | This study |
| JJD251 | *E. coli* MC4100 ($\lambda$ cI857 $S_{105}$/A99V) | Wang (6) |

[a]Coli Genetic Stock Center.

reaching a critical threshold concentration in the membrane. On the other hand, burst size estimates depend on the production of viable virions, which is the culmination of a phage assembly process involving the concerted production of many more phage proteins that are expressed at different stoichiometries. Although cellular capacities affect both processes, the higher cell-to-cell variation observed in burst size estimates most likely arose from the cumulative effects of gene expression and the ensuing complex molecular assembly.

A high degree of variation in burst size at different lysis times may have consequences for phage $\lambda$'s fitness. Phage fitness is highly correlated with the host physiological state, which in turn is dependent on the environment. In studies that have used one-step growth curves, it has been observed that burst size increases and/or lysis time shortens as the physiological state of the host improves (9, 42–49). Such short-term changes in the values of phage traits triggered by changes in the host are described as viral phenotypic plasticity (13, 50–52). A recent theoretical approach suggested that burst size plasticity drives ecological and evolutionary dynamics by strengthening dynamic feedbacks between a phage, its host, and the environment (53). It was hypothesized that plasticity in burst size is more important than the plasticity in lysis time, especially under favorable growth conditions, which allows production of more virions within a shorter lytic cycle. Plasticity provides phenotypic diversity in a phage population and may render sensitivity to the host environment without the need for genetic changes. Our findings have revealed a high degree of variation in burst size, which conforms to these theoretical predictions.

## MATERIALS AND METHODS

**Bacterial strains.** The bacterial strains used in our study are summarized in Table 1.

**Plaque assays.** To obtain plaques of lysis-deficient $\lambda$ phages, an *E. coli* strain with the pS$_{wt}$ (6) plasmid, which expresses holin in *trans*, was constructed. This *E. coli* strain was grown overnight in TB broth (5 g NaCl and 10 g tryptone in 1 liter of water) containing 0.2% maltose and ampicillin (100 $\mu$g/mL) at 37°C. The overnight culture was diluted with an equal volume of TB plus maltose plus ampicillin and grown for another 2 h. A 100-$\mu$L aliquot of these cells was mixed with phage lysate and incubated at room temperature for 20 min to allow preadsorption. This mixture was then added to 3 mL of molten H-top agar (54), gently vortexed, and overlaid onto freshly prepared plates containing 35 mL LB agar. The plates were then incubated at 37°C and plaques were counted after 18 to 22 h. For each time point, single-cell burst sizes from ≈100 cells were estimated.

**Thermal induction and lysis of single cells.** Cultures of the lysis-deficient lysogen [*E. coli* MC4100 ($\lambda$ cI857 $S_{am}7$) $\lambda$III] were grown overnight at 30°C in LB broth supplemented with 0.2% glucose (LBG). Overnight cultures were diluted 100-fold in LBG and grown in a 30°C shaking incubator (200 rpm) until the cultures reached an optical density at 600 nm of 0.3 to 0.4. The cultures were transferred to a 42°C shaking water bath for 20 min to induce lysis. Induced cultures were quickly diluted in LBG (prewarmed at 42°C), and 200-$\mu$L aliquots were transferred into wells of a 96-well plate such that each well received on average 0.25 cells/well. This degree of dilution results in mostly empty wells and only 2% of wells will contain more than 1 cell (16). The plate was quickly transferred into a prewarmed plate reader (Tecan Infinite M200Pro) and incubated at 37°C with constant agitation (orbital shaking, amplitude 6, frequency 141.9 rpm). After incubation in the plate reader for the required time, a multichannel pipette was used to quickly transfer 100 $\mu$L of chloroform into each well. The plate was then shaken at room temperature for 10 min. This treatment ensured lysis of cells to release phage virions. Aliquots of 100 $\mu$L of the supernatant aqueous layer were carefully collected and used in plaque assays to enumerate phages. For the lysogen with functional holin, 1 hour of incubation was sufficient for natural lysis.

**Data availability.** Single-cell burst size estimates are available at Data Dryad (https://datadryad.org/stash/dataset/doi:10.5061/dryad.x3ffbg7pb).

## SUPPLEMENTAL MATERIAL

Supplemental material is available online only.

**SUPPLEMENTAL FILE 1**, PDF file, 0.3 MB.

## ACKNOWLEDGMENTS

This work was made possible by grant number 1R01GM124446-01 from the National Institutes of Health. We express much appreciation to Khem Ghusinga, Cesar Vargas-Garcia, Fabrizio Spagnolo, Ing-Nang Wang, Daniel Weinreich, and Ry Young for illuminating discussions regarding bacteriophage life history. Additionally, we are grateful to Ing-Nang Wang and Ry Young for sharing phage and bacterial strains.

Conceptualization, S.K., A.S., and J.J.D.; Methodology, S.K., A.S., and J.J.D.; Formal Analysis, S.K., A.S., and J.J.D.; Investigation, S.K.; Resources, A.S. and J.J.D.; Data Curation, S.K., A.S., and J.J.D.; Writing — Original Draft, S.K., A.S., and J.J.D.; Writing — Review & Editing, S.K., A.S., and J.J.D.; Supervision, A.S. and J.J.D.; Project Administration, A.S. and J.J.D.; Funding Acquisition, A.S. and J.J.D.

We declare that no financial or nonfinancial competing interests exist.

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
