## [Reviewer comments · Microbiology Spectrum]

Microbiology Spectrum

A Single-Cell Approach Reveals Intercellular Heterogeneity in Phage Production Capacity

Sherin Kannoly, Gabriella Oken, Jonathan Shadan, David Musheyev, Kevin Singh, Abhyudai Singh, and John Dennehy

Corresponding Author(s): John Dennehy, Queens College Department of Biology

Review Timeline:

Submission Date:	October 31, 2022
Editorial Decision:	November 28, 2022
Revision Received:	November 30, 2022
Accepted:	December 2, 2022

Editor: Olaya Rendueles Garcia

Reviewer(s): The reviewers have opted to remain anonymous.

Transaction Report:

DOI: <https://doi.org/10.1128/spectrum.02663-21>

November 28, 2022

Prof. John J. Dennehy
Queens College CUNY
Biology Department
65-30 Kissena Blvd
Queens, NY 11367

Re: Spectrum02663-21 (A Single-Cell Approach Reveals Intercellular Heterogeneity in Phage Production Capacity)

Dear Prof. John J. Dennehy:

Thank you for submitting your manuscript to Microbiology Spectrum. As you will see your paper is very close to acceptance.

Please revise the Data availability statement. Note that your data should be publicly available to the community. We encourage you to upload this data to a repository, or added to the manuscript as a supplemental table, or extended dataset.

Please modify the manuscript along the lines I have recommended. As these revisions are quite minor, I expect that you should be able to turn in the revised paper in less than 30 days, if not sooner. If your manuscript was reviewed, you will find the reviewers' comments below.

When submitting the revised version of your paper, please provide (1) point-by-point responses to the issues raised by the reviewers as file type "Response to Reviewers," not in your cover letter, and (2) a PDF file that indicates the changes from the original submission (by highlighting or underlining the changes) as file type "Marked Up Manuscript - For Review Only". Please use this link to submit your revised manuscript. Detailed instructions on submitting your revised paper are below.

Link Not Available

Sincerely,

Olaya Rendueles Garcia

Reviewer comments:

Reviewer #1 (Comments for the Author):

This is a very nice study of burst-size variation in phage lambda. The work has been done carefully and the paper is written very clearly. I only have a few brief comments.

- In several places you write "each well contained 0.25 cells" or similar. I think you mean "on average", and you should state so. Otherwise I'm wondering whether you somehow split the cells into pieces and distributed those among wells.

- Is it possible to model the burst size distribution as a mixture of two distributions, one for single cells and one for double cells? This may be too ambitious, and I don't want to express that this is necessary for acceptance of the paper, but it's something to think about.

- I think you should make your burst size measurements available in some form, either deposit in an appropriate archive or provide alongside the paper as supplementary material. "available on request" has many many problems, even if all participating parties are operating in good faith (for example, authors die, data sets get lost, etc.), and should no longer be relied upon in my opinion.

Preparing Revision Guidelines

Please return the manuscript within 60 days; if you cannot complete the modification within this time period, please contact me. If you do not wish to modify the manuscript and prefer to submit it to another journal, please notify me of your decision immediately so that the manuscript may be formally withdrawn from consideration by Microbiology Spectrum.

Response to Reviewer Comments

Reviewer #1 (Comments for the Author):

This is a very nice study of burst-size variation in phage lambda. The work has been done carefully and the paper is written very clearly. I only have a few brief comments.

We thank the reviewer for their kind words.

- In several places you write "each well contained 0.25 cells" or similar. I think you mean "on average", and you should state so. Otherwise I'm wondering whether you somehow split the cells into pieces and distributed those among wells.

We appreciate this suggestion and added "on average" to each place where 0.25 cells per well is mentioned.

- Is it possible to model the burst size distribution as a mixture of two distributions, one for single cells and one for double cells? This may be too ambitious, and I don't want to express that this is necessary for acceptance of the paper, but it's something to think about.

It would be possible, but frankly none of the authors wanted to make the effort to do this unfortunately. Our intention instead is to repeat these measurements using a newly acquired FACS and compare the resulting data.

- I think you should make your burst size measurements available in some form, either deposit in an appropriate archive or provide alongside the paper as supplementary material. "available on request" has many many problems, even if all participating parties are operating in good faith (for example, authors die, data sets get lost, etc.), and should no longer be relied upon in my opinion.

Very true. We uploaded the data to Data Dryad.

December 2, 2022

Prof. John J. Dennehy
Queens College Department of Biology
Biology Department
65-30 Kissena Blvd
Queens, NY 11367

Re: Spectrum02663-21R1 (A Single-Cell Approach Reveals Intercellular Heterogeneity in Phage Production Capacity)

Dear Prof. John J. Dennehy:

Your manuscript has been accepted. Please make sure the data is accessible on DataDryad after publication. I am forwarding it to the ASM Journals Department for publication. You will be notified when your proofs are ready to be viewed.

Sincerely,

Olaya Rendueles Garcia
Editor, Microbiology Spectrum
